# Innovation and cumulative culture through tweaks and leaps in online programming contests

Elena Miu [1], Ned Gulley[2], Kevin N. Laland[1] & Luke Rendell [1]

The ability to build progressively on the achievements of earlier generations is central to human uniqueness, but experimental investigations of this cumulative cultural evolution lack real-world complexity. Here, we studied the dynamics of cumulative culture using a large-scale data set from online collaborative programming competitions run over 14 years. We show that, within each contest population, performance increases over time through frequent 'tweaks' of the current best entry and rare innovative 'leaps' (successful tweak:leap ratio = 16:1), the latter associated with substantially greater variance in performance. Cumulative cultural evolution reduces technological diversity over time, as populations focus on refining high-performance solutions. While individual entries borrow from few sources, iterative copying allows populations to integrate ideas from many sources, demonstrating a new form of collective intelligence. Our results imply that maximising technological progress requires accepting high levels of failure.

---

[1] Centre for Social Learning and Cognitive Evolution, School of Biology, University of St Andrews, Harold Mitchell Building, St Andrews KY16 9TH, UK. [2] MathWorks, 3 Apple Hill Drive, Natick, MA 01760, USA. Correspondence and requests for materials should be addressed to E.M. (email: elena.miu@gmail.com)

The cultural transmission of knowledge and skills, and concomitant iterative advances in technology, have led to extraordinary demographic and ecological accomplishments in our species[1], but the mechanisms that give human culture its potency remain unresolved. 'Cumulative culture'—the build-up of learned knowledge over time—allows populations to construct incrementally improved solutions that could not have been invented by a single individual[2–4], often associated with an increase in efficiency, complexity, and diversity[5]. This unique human accumulation of knowledge is widely thought to rely on a set of cognitive processes that include teaching, language, imitation and prosociality[4,6–8], and seems to be facilitated by increased population size and connectivity[9–12], although the underlying causes are not well-understood.

Cumulative culture is a large-scale, and potentially long-term, population-level phenomenon, the experimental investigation of which presents distinctive methodological challenges. Although theoretical analyses imply that factors such as high-fidelity information transmission and large, well-connected groups are important[11–14], it has proven particularly difficult to acquire an experimental understanding of the phenomenon in contexts that approach real-life complexity. As a result, experimental approaches to-date have been restricted to simplified cases, such as the building of paper airplanes, or towers from spaghetti and plasticine[15,16]. Such studies, like others using simple tasks like building virtual fishing nets[17] or knot-making[18], are informative, but do not approach the intricacy and richness of real-world cumulative cultural evolution.

**Fig. 1** Scores over time. Normalised log-transformed scores over time (measured in days from the beginning of each contest) in all contests (n = 47,921 entries). For visualisation purposes, because some values were zero, we added a small number on the appropriate scale to each score before log-transforming (the number chosen was 10). Note that in all contests low score values are better. Each point on the graph is an entry. The red line follows the progress of the leading entries in the contest, i.e. the entries that achieved the best score at the time of their submission

Here we present a detailed investigation of cumulative cultural evolution in a large-scale context that reflects the real-world complexity of human behaviour. We analysed a database of 21,745,538 lines of computer code in total and 483,173 unique lines, originating from 47,967 entries to 19 online collaborative programming competitions organised over the course of 14 years by the MathWorks software company. In every contest, the organisers set a computational challenge and, over the course of one week, participants developed and provided solutions in the form of MATLAB® code. Once an entry had been successfully evaluated, its score, code and the username of the participant who submitted it became public and available to all the other participants to build upon. The challenges were all NP-complete computer science problems[19], which are defined as problems for which an algorithm could not have found exact solutions in the contest timescale for problems of sufficient size (see Supplementary Methods). This resulted in reliance on heuristic and approximate solutions, thus allowing for open-ended improvement in the task. Hence, each contest functioned as a microcosm of cumulative cultural evolution in the sense of measurable improvement in task and, when combined, provide a data set of unprecedented richness, scale and complexity. Here we show that, in this context, individuals rely heavily on copying the current best entry. Collective improvement is achieved through a combination of small modifications of the current solution and radical innovations, the latter much less common, riskier, yet potentially more rewarding. As a result, the population converges on similar solutions over time, which decreases cultural diversity. Concurrently, a new form of collective intelligence arises through the recombination of existing ideas.

## Results

**Cumulative improvement of solutions**. Consistent with the expectation of cumulative advancements, we found that scores steadily improved throughout all contests (with the best solution on average scoring 40 times better than the first, Fig. 1). Only a small proportion of entries improved upon the current leading score (c. 6%, Supplementary Fig. 1), so each contest posed a genuinely challenging task. While each contest had a unique history of improvement, across all contests we observed extensive variation in the rate of improvement over time, with distinct periods of stasis, punctuated by large jumps in score associated with distinctive coding innovations.

**Successful code is copied**. Participants copied each other extensively within contests: across all contests, all entries bar the first contained at least one line of code from a previous submission, and overall only 3.8% of the entries contained at least 50% novel lines (Supplementary Fig. 2). This convergence was not based on the structure of the programming language—the average number of lines shared between entries from separate contests was just 8 (±15), compared to an average number of lines shared between entries in the same contest of 200 (±350).

Analysis of within-contest code similarity (measured between pairs of entries using the Czekanowski similarity index—see Methods) revealed that copying was not indiscriminate as participants exhibited a strong preference for copying the current leader. The baseline distribution of similarity between entries at the beginning of the contests (when participants only have access to their own information) shows a clear skew towards zero (Fig. 2a; Supplementary Fig. 3), whilst when all entries are considered a very large proportion of entries exhibit strong similarities to the current leader (Fig. 2b). Overall, 50% of the entries had a similarity >0.9 to the current leader, but only 26% had such a similarity to entries submitted after the current leader,

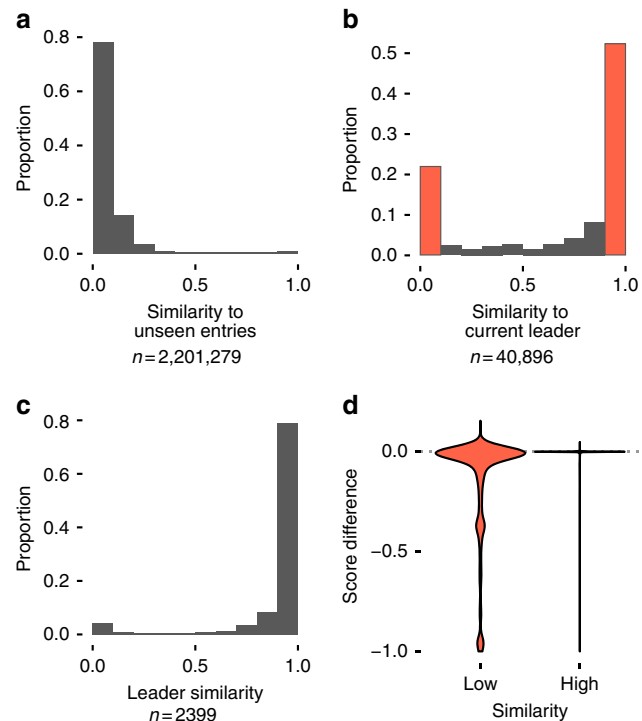

**Fig. 2** Similarities across all contests. Proportion of similarities **a** between all entries under limited visibility conditions (i.e. when players only had access to their own entries—see Supplementary Methods; n = 2,201,279); **b** of all entries to the current leader at the time when the entry was submitted (red indicates tweaks and leaps, with score difference plotted below; n = 40,896); **c** of that subset of entries that took the lead when entered to the previous leader (n = 2399). **d** Score difference for entries with low similarity (<0.1) and high similarity (>0.9) to the current leader—score differences for highly similar entries are narrowly distributed around 0, while low similarity entries show wide variance (n = 31,575)

confirming that it is the best-performing entry that is being copied. This strategy is even more strongly manifest in entries that took the lead when submitted (Fig. 2c), 91% of which had a similarity >0.5 to the previous leader. Relatively few new leaders introduced substantial amounts of novel code.

**Innovations include both 'tweaks' and 'leaps'**. The matrix of similarities between entries provides a graphical illustration of the pattern of copying throughout each contest, revealing how the submitted entries are grouped into clusters of similar solutions (Fig. 3). Analysis of the magnitude of change relative to the prior leader reveals a strongly bimodal distribution of improvements (Fig. 2c). We can characterise this distribution in terms of 'tweaks' and 'leaps': the population adopts a general solution for a while, incrementally improving it through modest refinements (i.e. 'tweaks'), but occasionally jumping to a new solution with low similarity to preceding entries (i.e. a 'leap').

**Leaps usually fail but can bring large advances**. The success of an entry—whether it took the lead, and if so by how much – was strongly related to the extent to which it was based largely on copying or exhibited substantial innovation. Among entries that took the lead, we observed a significant negative correlation between the entry's similarity to the previous leader and its associated improvement in score (Spearman's $\rho = -0.15$, $p < 0.001$), with the biggest improvements associated with those

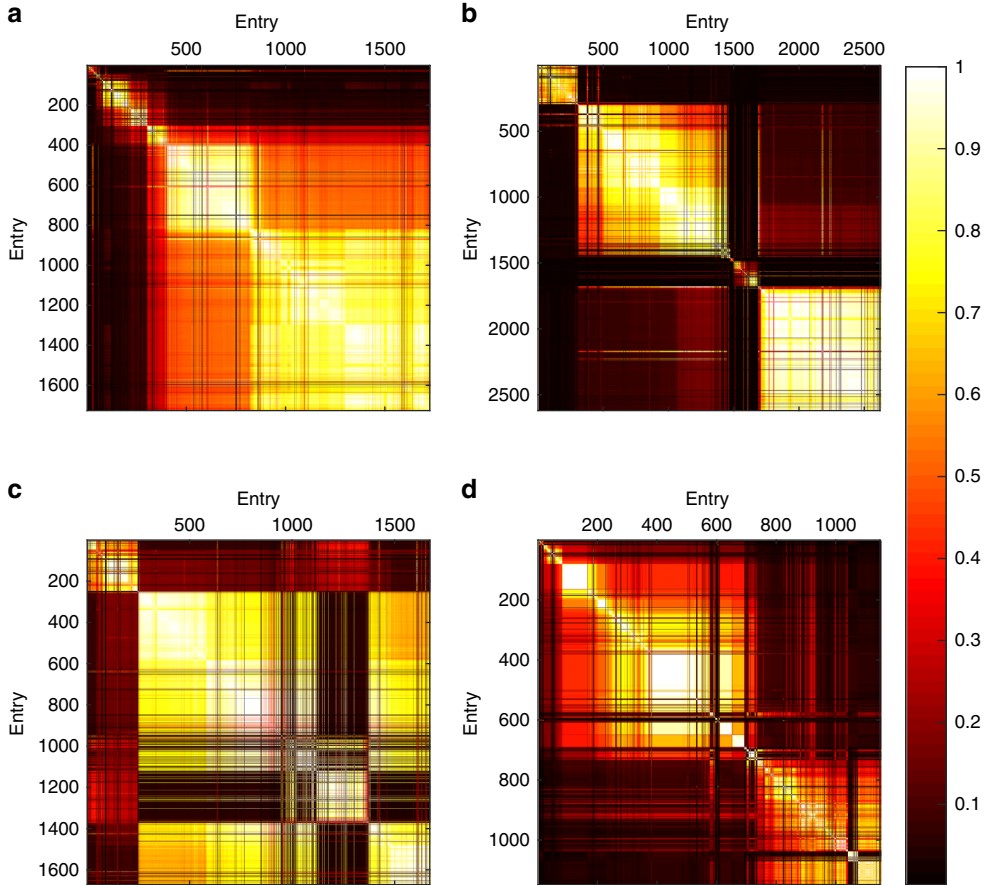

**Fig. 3** Code similarity matrices between all pairs of entries in four sample contests. Entries are ordered chronologically on time of entry from top to bottom and left to right. Each point represents the similarity between two entries. The bright squares show series of entries similar to each other compared to previous and subsequent periods, representing minor variations on a solution 'theme'. The bottom-right corners of the bright squares show where the population switched to a new form of solution that was relatively dissimilar to those before, and the top-left corners show where a new solution 'theme' began. Data are shown for four contests: **a** Gerrymandering—April 2004 ($n = 1726$), **b** Peg Solitaire—May 2007 ($n = 2620$), **c** Colour Bridge—November 2009 ($n = 1671$) and **d** Mars Surveyor—June 1999 ($n = 1148$; for ease of visualisation, here we only plot the entries for which the participants reported copying—see Supplementary Methods). Tweaks are manifest as lighter colours, and leaps as darker regions. **a** The final large cluster of solutions shows intermediate similarity to a previous smaller group. **b** The three distinct groups of similar solutions show the population switching to completely new ideas. **c** The population reverts to a previous idea, generating high similarity between the final and second group. **d** A contest characterised by rapid shifts in ideas

entries most different from the previous leader. However, among entries that did not take the lead, the reverse relationship was observed (Spearman $\rho = -0.53$, $p < 0.001$), with the most innovative entries exhibiting the poorest performance, measured as the absolute difference in score from the current leader. Hence tweaks were associated with smaller changes in score, either positive or negative, while leaps garnered both large improvements in score and spectacular failures (Fig. 2d; Supplementary Fig. 4). The distribution of entry performance relative to the current leader shows that while leaps were more likely to lead to poorer scores than tweaks of copied material overall, on rare occasions they generated significantly larger benefits.

**Most advances are incremental tweaks**. These observations support the intuition that copying combined with a modest tweak is a relatively 'safe' strategy, whilst the innovation manifest in 'leaps' is risky. Here the 'risk' is in terms of time wasted developing a new solution under heavy time constraints. The safe choice is to tweak the current best entry, and this is what most participants do. Across all entries, we observed substantially more tweaks than leaps, with a ratio between 2:1 and 3:1, depending on

the methods deployed to characterise tweaks and leaps (Supplementary Methods), and with most entries exhibiting high similarity to the current leader. However, amongst successful entries that took the lead, the ratio of tweaks to leaps was 16:1, overwhelmingly larger than in all entries. After the first 1–2 days of a contest, leading solutions rapidly became difficult to beat because they have benefitted from the concerted efforts of multiple people, and it was only rarely that an innovative solution took the lead. Mean (±SD) increments in performance associated with successful tweaks and leaps, normalised across contests, were 0.0003 (±0.0022) and 0.0033 (±0.0171), respectively.

**Tweaking and copying reduce code diversity over time**. The strong preference for tweaking copied code generated population-level patterns of cultural diversity over time. Through copying, the population converged on similar solutions for substantial periods of time, causing a decrease in the diversity of entries, as measured by the total number of distinct lines of code present across the population of entries in a given time period (Fig. 4, Supplementary Fig. 5, 6). Bayesian mixed models with beta-distributed errors indicated an average change of −0.15 (95%

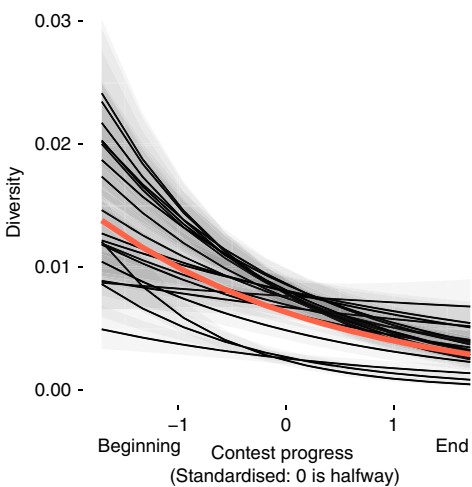

**Fig. 4** Cultural diversity decreased over time. Generalised linear mixed-model predictions of cultural diversity over time. Each black line plots the predictions for a single contest, whilst the red line plots the overall estimate. The shaded area indicates the 95% credible interval of the posterior estimates per contest ($n = 1845$ data points binned by time, see Methods)

credible interval: $-0.19$ to $-0.12$) in the logit of diversity for every additional 10% of contest entries (equivalent to a drop in normalised diversity from 0.01 to 0.086).

Copying also affects how individuals combine ideas from previous entries. Recombination of different ideas is widely thought to be a major driving factor in cumulative culture[20], but quantitative evidence is scarce. We defined 'recombination' strictly as the association of cultural traits already present in the population, and thereby distinguish recombination from the combining of new and old ideas ('refinement'). For each entry, we tracked each line of code back to the first entry that introduced it into the contest, and characterised each entry in terms of its number of such original sources. This allowed us to quantify how many original sources each entry drew from, and how long such sources persisted over time.

**Successful solutions combine earlier advances**. Entries drew on many tens, sometimes hundreds, of original sources overall (mean ± SD = 49 ± 38; Fig. 5a). Across all contests, the amount of recombination, measured as the number of original sources utilised, increased over time (Spearman $\rho = 0.53$, $p < 0.001$) and was both positively correlated with similarity to the current leader (Spearman $\rho = 0.44$, $p < 0.001$) and negatively correlated with the difference in score to the current leader (Spearman $\rho = -0.25$, $p < 0.001$), where being different over-whelmingly means a worse entry. Therefore, copying leads to an accumulation of recombination in the population—entries borrow from more and more original sources with time—and this accumulation of ideas is correlated with better performance (Supplementary Fig. 7). Moreover, successful tweaks drew on a larger number of original sources than successful leaps (mean ± SD = 64 ± 37, 36 ± 33, respectively), suggesting that copying was a key driver of population-level recombination (Supplementary Fig. 8).

We also estimated the immediate number of sources an individual entrant utilised by tracking each line of code to the most recent prior entry containing it. This is a conservative estimate that assumes copying is the dominant strategy, and that people submitting code tended to copy the latest entries. We found that 90% of all entries drew on five immediate sources or

fewer (mean ± SD = 2.82 ± 2; 25% refined only a single entry), with no increase in the number of recent sources utilised over time (Fig. 5b, c). In contrast to the use of original sources, successful leaps drew on larger numbers of immediate sources than successful tweaks (mean ± SD = 3 ± 2, 2 ± 1, respectively), suggesting that these leaps were often associated with repurposing of old code. These findings show that populations are able to combine and synthesise a very large (up to tens or hundreds) and increasing number of initial sources, which persist over time as a result of the high levels of copying, despite the fact that each individual entrant themselves draws on a small number of sources. This is further emphasised by the fact that successful entries draw on original sources from on average 19 (±9 SD) different individual contestants. In this light, cumulative culture can be seen to exemplify a novel form of collective intelligence[21–23].

## Discussion

The dominant strategy exhibited by entrants (i.e. copying the most successful entry) is known as 'success-', or 'payoff-', 'biased' social learning[24] and it is greatly facilitated in these contests by explicit payoff information. Yet the observed convergence of the population over time on similar behaviour leads to population-level patterns comparable to expectations under 'conformist transmission', where individual disproportionately adopt the majority behaviour[4]. There is much current debate regarding the degree to which individuals conform, and the contexts in which conformity bias is adaptive[25,26]. Theoretical analyses demonstrate that conformity can have substantial fitness benefits, particularly in spatially variable environments and that, by maintaining stable between-group differences, conformity can support a cultural group selection explanation for the evolution of cooperation[4,27]. By showing that cumulative culture generates a conformity-mimicking homogeneity in behaviour without an explicit conformist bias, and thereby broadening the opportunity for selection between groups to arise, our study sheds new light on the roles that success-biased copying and cultural group selection might play in cultural evolution. Furthermore, once early human culture became cumulative, the mechanism we have illustrated here could have led to behavioural convergence that then fed back to select for conformity precisely because, as we have also shown, cumulative culture makes copying a low-risk option[28].

The observation that cumulative culture can reduce cultural diversity may appear paradoxical given the widespread view that cumulative culture generates diversity. Here our study offers a further novel insight. Recent analyses of human innovation have distinguished between two classes of innovation, namely 'better-faster-cheaper' solutions and innovative solutions that generate a cascade of new possibilities and thereby deliver entirely new kinds of functionality[29]. New functionality was precluded in the MathWorks contests, which raises the possibility that the diversity of human culture does not arise from cumulative cultural processes per se, but rather from that subset of innovation that affords novel functionality. Additional cultural diversity may emerge from the context specificity observed across entire populations, including that afforded by other cultural knowledge, as well as from ecological variation, which can provide new problems for cumulative culture to solve.

The tweak-and-leap nature of cultural evolution revealed here has parallels with both scientific revolutions[30], though in our case leaps are restricted to improvement in a single domain, and the idea of punctuated equilibrium[31–33], although it is debatable whether tweaks truly reflect periods of stasis, given the amount of improvement they deliver. Our study also has strong implications

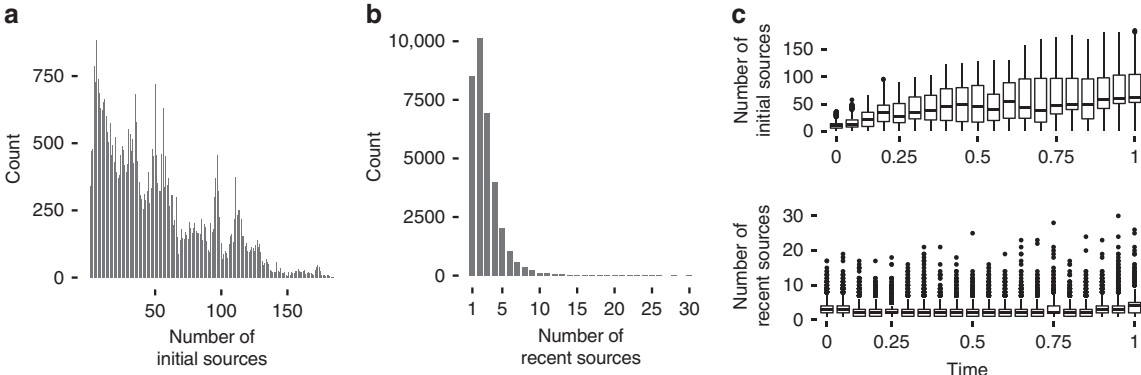

**Fig. 5** Recombination over time. **a** Number of initial sources (n = 42,072—all entries excluding those that do not use recombination) and **b** number of recent sources for all the entries in all contests (n = 37,474—all entries that use recombination, excluding the 'Blackbox' contest, in which one single participant submitted a disproportionate amount of entries refining their own solution, skewing the data); **c** number of initial and recent sources over time, averaging across all contests (n = 37,474)

concerning the way human groups achieve collective improvements in performance and thereby drive technological progress. Medical researchers analysing the use of coding challenges, similar to the contests we analysed, to crowd-source models of cancer prognosis have warned against the problem of low solution diversity[34], and our analysis reinforces these concerns. Other research has shown that novelty in science is risky, but can be associated with considerably higher long-term impact[35], and our analysis endorses these findings. We note that in the contests populations do not necessarily converge on the globally best solution. Individuals commonly and rationally capitalise on the investment that the population has already made by tweaking the current best entry, but this 'safe' approach tends to result in the neglect of new innovations, and/or a failure to realise their potential, if innovations do not immediately become market leaders. The danger here is that individuals may pursue the current consensus largely because of the effort that has already been put into it, analogous to the 'Concorde fallacy'[36], and that the manifest tendency to copy current leaders may encourage populations to pursue suboptimal solutions. A clear implication of our analysis is that the substantial leaps in progress that break this dynamic go hand-in-hand with a high risk of failure, and that institutions and organisations that seek genuinely revolutionary innovation must be prepared to invest in highly original, high-risk-high-return research whilst tolerating lower rates of success and substantial amounts of sometimes spectacular failure.

## Methods

**Data source.** We analysed data from 19 online programming competitions organised by MathWorks from 1998 until 2012. Participants could submit as many solutions as they wanted though an online interface, identified by a self-chosen username. Once an entry had been evaluated, its score, code and the username of the participant who submitted it became public and available to all the other participants. Although submission was completely anonymised, participants generally submitted solutions from a single ID, because the main motivation to win the contest was reputational—actual prizes were nominal (e.g. a branded T-shirt). The contest winner was the entrant with the best score at the end of the week. There were a number of variations on this basic model. In 2004, the organisers introduced two more stages to the competition. While the 6 previous contests allowed participants to view other entries from the beginning, the subsequent 13 contests used a framework consisting of three stages. On the first day, termed 'darkness', participants did not have any information on the scores of their own entries or other players'. During the second day, labelled 'twilight', participants only had information on the rank of their own entries compared to everyone else's. From the third day onwards, in 'daylight', they had full access to information concerning their own entries and the entries of other players, including their score, rank, and computer code. When analysing the data from these later contests, we only included data from the daylight condition—the only condition directly comparable

with earlier contests—while statistically controlling for this difference using mixed effects models.

Entries in the contests were scored by the MathWorks organisers as a function of their effectiveness on the task, the speed of execution, and code complexity, measured using McCabe's cyclomatic complexity[37], which takes into account the code structure by measuring the number of independent paths through a programme's source code. The first two factors weighed more heavily towards the final score, while limiting complexity ensured that entries remain concise so as not to lock up the computer evaluating the entries. Thus, improving task result score, or the speed of execution, or both, could all result in better contest scores. Entries were disqualified if they exceeded execution time or length limits.

Our sample included 19 contests, with an average of 2396 entries per contest (ranging from 1138 to 6367 entries) and an average 120 participants per contest (between 63 and 202). This amounts to a total of 47,967 entries containing a total of 483,173 unique lines of code (between 1 and 10,061 lines per entry).

**Measuring similarity between entries.** We used a variation of the Sørensen–Dice coefficient to measure the similarity between two entries[38,39]. Originally a statistic designed for comparing two ecological samples in terms of presence/absence of species, it has an extension, the Czekanowski similarity[40], which we use here, given by:

$$CZ_{ik} = 2 \frac{\sum_{j=1}^{S} \min(x_{ij}, x_{kj})}{\sum_{j=1}^{S} (x_{ij} + x_{kj})}, \quad (1)$$

where $CZ_{ik}$ is the similarity between samples $i$ and $k$, $x_{ij}$ is the number of instances of species $j$ in sample $i$, $x_{kj}$ is the number of instances of species $j$ in sample $k$, and $S$ is the total numbers of species. For our analysis, each sample corresponds to an entry, and each species is a line of code—two lines of code are considered identical when they both consist of the same set of characters (excluding spacing and capitalisation) in the same order. Every entry is a set of lines of code, so the similarity between two entries is a function of the total number of lines they have in common, including reoccurring lines, relative to the sum of their lengths.

The Czekanowski similarity does not take context into account—it relies on the number of lines, disregarding order and potential sequences of lines that might reappear together—yet it performs very well for our purposes. If two entries have a high Czekanowski similarity, they are almost certainly similar in terms of the order of lines of code, as it is extremely unlikely that the same lines could be combined differently in a piece of code that remains functional. Moreover, the nature of computer code ensures that the lines in an entry will be highly idiosyncratic because variable names are arbitrarily selected from a vast sample space (MATLAB variable names can be up to 63 characters long), so the chance of finding the exact same lines in two entries that are not actually functionally identical is very low. Thus the Czekanowski similarity between two unrelated entries will be low, while two functionally related entries would exhibit high Czekanowski similarity.

**Analysing cultural diversity.** To measure diversity, we needed to group entries to create snapshots of the 'culture' of the population at given points in the contest. We sorted entries by order of submission and grouped them in bins each containing 1% of the entries in the contest, and then computed a diversity measure for each bin. Our measure over the contest reflects the diversity of the first 1% of the entries, the following 1%, and so on. Within each bin we measured diversity as the number of unique lines of code entered divided by the total number of lines submitted in the contest (Supplementary Fig. 5). We used bin number as a measure of progress

through the contest since the entries were ordered chronologically, and we were not interested in the absolute time of submission but in how diversity changed over the course of the contest, independently of the varying rate at which entries were submitted.

We measured the increments in performance for each entry as the absolute difference score between each entry and the current leader at the time of its submission, and normalised across contests using: $\bar{x} = \frac{x - \min(x)}{\max(x) - \min(x)}$, where $\bar{x}$ is the normalised, and $x$ is the un-normalised increment value.

**Statistical analysis**. We fitted a Bayesian mixed-effect regression model using Monte Carlo Markov Chain (MCMC) methods using the Bayesian fitting program JAGS[41] via the r2jags package in the R[42] software. As diversity is measured as a proportion between 0 and 1, we used beta-distributed errors, with a logit link. We used diversity as the dependent variable, and chronological bin number as a fixed effect. Each contest proposed a different problem and had a unique set of participants, which meant that entries from within the same contest could not be considered independent. To account for this, we included normally distributed random effects, allowing both the intercept and the slopes in the diversity–time relationship to vary with contest. The model definition, parameterised as per ref. [43] is given below:

$$Y_{ij} \sim \mathrm{beta}\left(\mu_{ij}, \phi\right); E\left(Y_{ij}\right) = \mu_{ij}; \mathrm{var}\left(Y_{ij}\right) = \frac{\mu_{ij} \times (1 - \mu_{ij})}{(\phi + 1)},$$
$$\mathrm{logit}\left(\mu_{ij}\right) = \alpha_j + \beta_j \mathrm{Bin}_{ij}; \alpha_j \sim N(\mu_{\mathrm{int}}, \sigma_{\mathrm{int}}^2); \beta_j \sim N(\mu_{\mathrm{slope}}, \sigma_{\mathrm{slope}}^2).$$

We present results from an MCMC run of 50,000 iterations after burn-in, with 3 chains, and a thinning value of 10, resulting in 5000 posterior samples, with adequate mixing. We used uninformative Gaussian priors for the $\mu$ estimates, and flat uniform priors for $\phi$ and each $\sigma$.

**Ethical statement**. This work was approved by the University Teaching and Research Ethics Committee of the University of St Andrews (approval code BL11221).

**Data availability**. The data and the code used for analyses are available at [https://osf.io/xvtuy/?view_only=48ae607af38249cdb59965d2f11175b6].

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

## Acknowledgements

E.M. was supported by the John Templeton Foundation Grant #40128 'Exploring the Evolutionary Foundations of Cultural Complexity, Creativity, and Trust' and the University of St Andrews School of Biology. L.R. was supported by the Marine Alliance for Science and Technology for Scotland (MASTs) pooling initiative funded by the Scottish Funding Council (grant reference HR09011).

## Author contributions

E.M., L.R. and N.G. designed the study. N.G. performed the data collection, and E.M. analysed the results. E.M., L.R. and K.N.L. prepared the manuscript.

## Additional information

**Competing interests:** The authors declare no competing interests.

