## [Peer Review File · Nature Communications]

Reviewers' comments:

Reviewer #1 (Remarks to the Author):

This manuscript reports on the analysis of the evolution of computer code submitted to 19 online contests run by Math Works comprising hundreds of thousands of lines of code and tens of thousands of entries. The aim of the paper is to test the idea that humans take advantage of social learning to create a sort of simulation of organic evolution that leads to the cumulative evolution of cultural adaptations. In the contests, participants had access to the code submitted by their competitors and data on the success of that code. The contests continued for about a week and participants could modify their code and submit successive modifications for renewed evaluation, incorporating as much or as little code from others' submissions as they wanted.

Heretofore, attempts to formally test the importance cumulative culture have relied on mathematical models, short-term ultra-simplified experiments (all that can be accomplished with the resources normally available to researchers), or arguments based on historical cases that lack much detail on the underlying cultural evolutionary processes. In most real-world situations, it is quite hard to get information on who is influenced by whom, for example. We can trace what is evolving but typically can't convincingly argue why rates and directions of change are as they are. This study takes advantage of a real-world example of cultural microevolution where a large amount to rather detailed data documents the underlying processes, leaving little ambiguity about mechanistic details.

The headline result of the study is that most modifications to programs in the course of a contest were "tweaks," small changes to the code, while a small minority were "leaps" in which a major conceptual change in a program was attempted. Tweaks were much more likely than leaps to make an improvement in performance of the code, but leaps that did succeed typically resulted in larger performance improvements than tweaks.

There were other patterns that were interesting. Unsurprisingly, contestants frequently adopted changes from programs that were successful in the previous round. This led to a decline in the diversity of code as time went on. Individual contestants borrowed from relatively few other programs but at the population level recombination was quite efficient leading most submissions to improve markedly over time due to the spread of successful innovations. Typically, programs at the end of the contest benefited from many tweaks and a few leaps.

These results rather strongly support the idea that cumulative culture is important. Prominent skeptics such as Steven Pinker and Olivier Morin have argued quite vehemently against the idea that cumulative culture is important in humans. I believe that this study is the single best investigation supporting the importance of cumulative culture in our species, strongly buttressing arguments based on models and weaker data. I hope it will inspire others to hunt for similar special cases where the processes of cultural evolution can be explored in such high resolution.

Reviewer #2 (Remarks to the Author):

In this manuscript the authors analyze a recent series of programming competitions to extract quantitative clues and qualitative patterns of cultural transmission and cumulative culture evolution. The data used by the authors was massive (~0.5M lines of code in ~50K entries across 19 contests) and contained some of the real-life complexities lacking in previous oversimplified experimental approaches of the past. The contests were organized by the software company MathWorks to promote the development and sharing of Matlab code. In a commendable exercise of thinking outside of the box, the authors used what could be considered a dry computer science experiment to shed light on the evolution of cultural trends and the nature of cultural

diversity and innovation.

There is a rich set of results emerging from this paper, most of which are relatively intuitive and in hindsight expected. For example, the diversity of solutions declined in time, as solutions clustered around the proven best entry so far. The contents of each code was an aggregate of the history of the contest, even though each entry drew from a handful of sources to copy from, which the authors call a new form of collective intelligence. The winning entries at any time point were based on tweaks over the previous best, but there were occasional leaps that disrupted the predominant culture and added an innovative component that was then copied and maintained by tweaks until the next leap. Most leaps led to failure, but the risk to fail was paid off by the fact that without leaps there was no disruptive success. One of the interesting thing in this paper is the ability of the authors to quantify these trends. For example, the rate at which success was obtained by a tweak versus a leap was 16:1. However the mean increment in performance by a tweak is ~ 2 while for a leap is ~ 24 .

There are passages of the manuscript that need some clarification. Here are a few:

- 1) Figure 1. If log in the y-axis is to the base e, the average improvement seems to be more than the reported 40, when according to the authors the best solution on average scores 40 times better than the first. Please explain.
- 2) In Supplementary Figure 1, add the time scale in the x-axis.
- 3) In Supplementary Figure 2, what is box in subfigure B represent? It is unclear whether it indicates a box and whiskers or something else. Please explain.
- 4) In line 261 and 262, where it says: "S is the total numbers of samples", I believe it should be : "S is the total numbers of species",

In all, I find that the ideas of the paper are original, the manuscript is well written, and the methods are sound. There are, however, a few questions and topics I would like to see further discussed. Below are some ideas for the authors consideration.

- 1) How many of the tweaks are bug fixes of the same users on their code?
- 2) What are the implications that the objective criterion of performance in the competition is explicitly spelled out, on the papers' conclusions about how culture is transmitted and accumulated. For example, the authors mention that (line 172) "conformity can have substantial fitness benefits...", but in the case of the paper, fitness (that is, the performance of an entry) fosters substantial conformity benefits (tweaks). In the real world (as opposed to a well-controlled competition) the fitness or objective criteria of success are much more subjective and multidimensional. My intuition is that this limits the generalizability of the conclusions of the paper to special cases of cultural evolution, such as scientific progress.
- 3) About the previous point, it seems that the conclusions of the paper could have interesting implications in epistemology, and in particular to inform Kuhn's ideas in "The structure of scientific revolutions". The analogy would be that tweak epochs are equivalent to the relatively quiescent time of paradigm consolidation and constructive growth of the scientific corpus while the leaps are the times of challenging the paradigm and disruptive thinking. In science, the leaps come typically from some experiments that contradict the reigning paradigm. Is there an equivalent of a need for a leap in technology, or in these competitions?
- 4) Do the 'good' leaps always come from the same participants?
- 5) Some of the ideas proposed by the authors have been observed in similar settings. For example in Margolin et al: Systematic Analysis of Challenge-Driven Improvements in Molecular Prognostic Models for Breast Cancer, Science Translational Medicine, Vol. 5, Issue 181, pp. 181re1 (2013), the authors write: "Finally, the sharing of ideas enabled by requiring submissions as re-runnable source code may ironically inhibit the diversity of innovations, effectively encouraging a monoculture as the community converges on a local optimum, modifying and extending approaches with high performance in the early stages of feedback". The authors conclude that "it is important to develop a reward system that favors collaborative research practices that balance

the currently prevalent winner-takes-all reward system." I wonder if the authors have any wisdom to share about this.

6) About the previous point, there is a need to distinguish between copying code and collaboration. The reduction of diversity and approach to local minima during the tweak phases are still done in framework that is more competitive than collaborative. How would this change if there was a collaborative period at the end of each competition where the right incentive would motivate the group to collaborate and not to copy? For example, in many of the recent DREAM Challenges, a competitive phase (equivalent to a long twilight zone) is followed by a collaborative phase (see., eg. DOI: 10.1038/ncomms12460 or <https://www.synapse.org/#!Synapse:syn4224222/wiki/401760>).

7) Community Challenges have become an alternative and popular way to do scientific research. However, there is not too much research on how to optimize these Challenges for maximal gain, and I think the authors' work could be a valuable contribution to this literature. Can the authors suggest how the results presented in their paper can be used to improve scientific or coding competitions? Is there any way to associate the fitness (score) to the incentives that would increase the number of leaps over tweaks? For example, there is a way to report results in a leaderboard called "LADDER" that only reports the new result if it is considerably (statistically significantly) different than the previous best submission (see <https://arxiv.org/abs/1502.04585>). Maybe if the code is shared only when the new best score is posted there would be more time for independent tinkering. Would this be a good compromise between not sharing and whole sharing of source code?

8) The competitions described in the paper use a completely open framework to foster innovation, and it is one that, as the authors show, progresses by tweak and leaps. At the same time, there is a body of literature in machine learning on ensembles of algorithms. These ensemble methods require independent thinking, and the aggregation of independent weakly predictive solutions can yield a large benefit, often better than the best of all solutions (see, e.g., Nature Methods volume 9, pages 796-804 (2012)). Is there a good compromise between these two way of thinking: aggregating independent solutions to create an ensemble that perform extremely well in spite of the components being weak predictions, versus correlated solutions that tweaks to improve until the next better and independent solution shows up?

Reviewer #3 (Remarks to the Author):

The authors analyzed a rich dataset about MATLAB programming contests that have implications far beyond computer programming. These contests are interesting in that participants are allowed to see others' submissions and build upon them, which allows for the study of collective intelligence and group problem-solving. The key results of this paper include:

- 'Tweaks' in performance occur frequently while performance 'leaps' are rare. Furthermore, the variance in the change in solution's performance is much higher for 'leaps' than for 'tweaks'.
- Technological diversity reduces over time as innovators settle on high performing solutions.
- Evidence of collective intelligence emerges from sequences of entries built upon the current most promising solutions, which are aggregations of the most promising solutions from previous generations.

Comments:

1. Why were only 19 programming competitions used? There were more than 19 competitions and the author's online supplement includes data for 20 competitions. Why were the other competitions omitted from the analysis?
2. Line 27-28: The wording of this could be made clearer.
3. Line 67-68: You could discuss the idea of "punctuated equilibrium" from evolutionary biology (see [1]) which also has been used in technological innovation research from economics (see [2])

and [3]).

4. Line 69-70: "all entries bar the first contained at least one line of code from a previous submission", but in Figure 3 of the extended data there appears to be almost a third of the entries that have "zero new lines".
5. Line 73-74: Before the number of novel lines were reported in terms of percentages, then it changes to number of lines of code. It would be helpful to also include the percentage.
6. Line 83: Should it say "... after the current leader was no longer the leader"?
7. Line 100 and 103: Both correlation values are negative, but the paper states that the "reverse relationship was observed" for the second value. Are the values correct?
8. Line 121-122: Are the increments in performance normalized across competitions?
9. Line 126-127: Another analysis of the change in solution diversity in the MATLAB competitions showed that the diversity in the population of solutions dropped after a new leading entry [4].
10. Line 138: It would be good to know how many unique contestants contributed a line of code to the leading entries. This would ensure that the collective intelligence was coming from more than one contestant.
11. Line 262: Should this line be "the total number of species..."

By using a quantitative approach to show how recombination plays a role in collective intelligence, this paper is of interest to a broad audience of researchers in innovation studies and social learning. Moreover, the data used in this paper provides a unique view of the innovation process where all of the artifacts are available for inspection since they are preserved in digital form. There are other data sources where the changes in software are observable over time (e.g., data from the open source community), but none that I am aware allow for the study of collective intelligence of problem-solvers that have well-defined performance metrics while still maintaining a high level of problem difficulty.

[1] N. Eldredge and S. J. Gould, "Punctuated equilibria: An alternative to phyletic gradualism," *Production and Operations Management*, vol. 1, No. 4, pp. 334-357, 1992.

[2] J. Mokyr, "Punctuated Equilibria and Technological Progress," *American Economic Review*. 80. 350-54, 1990

[3] W. Abernathy and J. Utterback, "Patterns of industrial innovation," *Technology Review*, vol. 80, no. 7, pp. 40-47, 1978.

[4] R. Casstevens, "What leads to innovation: An analysis of collaborative problem-solving," 7th European Meeting on Applied Evolutionary Economics, Pisa, Italy, 2011.

Innovation and Cumulative Culture through ‘Tweaks’ and ‘Leaps’ in Online Programming Contests

Response to referees

We would like to thank the editor and the reviewers for all the helpful comments. Below we address each remark point by point (the comments are in bold).

Reviewer #1:

We are very grateful to the reviewer for their kind and supportive comments about the paper. No changes were requested.

Reviewer #2:

1) Figure 1. If log in the y-axis is to the base e, the average improvement seems to be more than the reported 40, when according to the authors the best solution on average scores 40 times better than the first. Please explain.

The reviewer is correct. For visualisation purposes, because some values were zero, we added a small number on the appropriate scale to each score before log-transforming (the number chosen was 10), which incidentally “stretches” the values close to the median, where the data displays the most overlap. We reported the raw values in the text, which explains the discrepancy. We have now clarified this in the figure legend.

2) In Supplementary Figure 1, add the time scale in the x-axis.

Time of submission added in the x-axis, measured as the number of days from the beginning of each contest.

3) In Supplementary Figure 2, what is box in subfigure B represent? It is unclear whether it indicates a box and whiskers or something else. Please explain.

They are standard boxplots – figure caption updated.

4) In line 261 and 262, where it says: "S is the total numbers of samples", I believe it should be : "S is the total numbers of species".

The reviewer is correct, line 262 updated (now line 275).

1) How many of the tweaks are bug fixes of the same users on their code?

Only code that produced a valid result was permitted as an entry, so any bugs that caused execution failures would not have entered the contest code base; in terms of ‘non-fatal’ bugs, distinguishing what is a ‘bug’ and what is sub-optimal code is not trivial. In some cases it might be possible to use human judgment to establish which entries could be users fixing bugs leading to poor

performance in their own code, but the volume of code we have available is so large that we do not have the logistical capacity analyse the code to this level.

2) What are the implications that the objective criterion of performance in the competition is explicitly spelled out, on the papers' conclusions about how culture is transmitted and accumulated. For example, the authors mention that (line 172) "conformity can have substantial fitness benefits...", but in the case of the paper, fitness (that is, the performance of an entry) fosters substantial conformity benefits (tweaks). In the real world (as opposed to a well-controlled competition) the fitness or objective criteria of success are much more subjective and multidimensional. My intuition is that this limits the generalizability of the conclusions of the paper to special cases of cultural evolution, such as scientific progress.

In this work we were explicitly concerned with cumulative cultural evolution in the sense of cultural evolution that produces measurable improvement in some objective (we have added clarification in line 61 The reviewer is entirely correct in pointing out that while in our data the payoffs are explicit and transparent, the real world offers many counter-examples, in which payoffs are opaque and/or implicit. Nonetheless, here we focus on trying to understand phenomena like technological progress, which are characterized by payoffs that are easy to read, such as increased fuel efficiency, for example (added clarification in the discussion, line 177).

3) About the previous point, it seems that the conclusions of the paper could have interesting implications in epistemology, and in particular to inform Kuhn's ideas in "The structure of scientific revolutions". The analogy would be that tweak epochs are equivalent to the relatively quiescent time of paradigm consolidation and constructive growth of the scientific corpus while the leaps are the times of challenging the paradigm and disruptive thinking. In science, the leaps come typically from some experiments that contradict the reigning paradigm. Is there an equivalent of a need for a leap in technology, or in these competitions?

This is a great suggestion and there are parallels that we now discuss in line 206. Our results suggest that innovatory leaps are a key part of cumulative culture.

4) Do the 'good' leaps always come from the same participants?

The question of individual variation in performance is entirely pertinent, and we are planning another analysis of the data to address this exact issue. However, including this analysis here would considerably increase the amount of material and exceed the limits of the journal format.

5) Some of the ideas proposed by the authors have been observed in similar settings. For example in Margolin et al: Systematic Analysis of Challenge-Driven Improvements in Molecular Prognostic Models for

Breast Cancer, Science Translational Medicine, Vol. 5, Issue 181, pp. 181re1 (2013), the authors write: "Finally, the sharing of ideas enabled by requiring submissions as re-runnable source code may ironically inhibit the diversity of innovations, effectively encouraging a monoculture as the community converges on a local optimum, modifying and extending approaches with high performance in the early stages of feedback". The authors conclude that "it is important to develop a reward system that favors collaborative research practices that balance the currently prevalent winner-takes-all reward system." I wonder if the authors have any wisdom to share about this.

We thank the reviewer for this helpful suggestion, which is now discussed in lines 212-214.

The reviewer also makes a series of related points that we discuss together below.

6) About the previous point, there is a need to distinguish between copying code and collaboration. The reduction of diversity and approach to local minima during the tweak phases are still done in framework that is more competitive than collaborative. How would this change if there was a collaborative period at the end of each competition where the right incentive would motivate the group to collaborate and not to copy? For example, in many of the recent DREAM Challenges, a competitive phase (equivalent to a long twilight zone) is followed by a collaborative phase (see., eg. DOI: 10.1038/ncomms12460 or <https://www.synapse.org/#!/Synapse:syn4224222/wiki/401760>).

7) Community Challenges have become an alternative and popular way to do scientific research. However, there is not too much research on how to optimize these Challenges for maximal gain, and I think the authors' work could be a valuable contribution to this literature. Can the authors suggest how the results presented in their paper can be used to improve scientific or coding competitions? Is there any way to associate the fitness (score) to the incentives that would increase the number of leaps over tweaks? For example, there is a way to report results in a leaderboard called "LADDER" that only reports the new result if it is considerably (statistically significantly) different than the previous best submission (see <https://arxiv.org/abs/1502.04585>). Maybe if the code is shared only when the new best score is posted there would be more time for independent tinkering. Would this be a good compromise between not sharing and whole sharing of source code?

8) The competitions described in the paper use a completely open framework to foster innovation, and it is one that, as the authors show, progresses by tweak and leaps. At the same time, there is a body of literature in machine learning on ensembles of algorithms. These ensemble methods require independent thinking, and the aggregation of independent weakly predictive solutions can yield a large benefit, often

better than the best of all solutions (see, e.g., Nature Methods volume 9, pages 796-804 (2012)). Is there a good compromise between these two way of thinking: aggregating independent solutions to create an ensemble that perform extremely well in spite of the components being weak predictions, versus correlated solutions that tweaks to improve until the next better and independent solution shows up?

Points 6-8:

The goal of this work was to understand human behaviour in a naturalistic cumulative cultural evolution scenario, and we acknowledge in the discussion that this behaviour could lead to suboptimal solutions when individuals are engaged in exploring a complex problem characterised by a rugged fitness landscape. Our goal was to describe how this process developed in an unconstrained context, rather than test and identify the optimal way to construct such challenges. Controlled experiments that would allow us to manipulate the reward structure and thereby affect the trade-off between collaboration and competition, as well as the visibility regimes that modulate the extent of convergence we observe, would certainly be beneficial in exploring potential optima for these types of problems, and indeed we are in the process of tackling some of these questions through experimental approaches that will allow us to answer these questions more confidently, but this is beyond the scope of this manuscript.

Reviewer #3:

Comments:

1. Why were only 19 programming competitions used? There were more than 19 competitions and the author's online supplement includes data for 20 competitions. Why were the other competitions omitted from the analysis?

20 contests were made available to us by our MathWorks collaborator. One contest (Army ants, Nov. 2008) was excluded because it followed a very different structure, according to which each day presented a different problem characterized by a different scoring scheme. This resulted in independent unrelated score improvement trajectories each day, which could not be compared with the other 19 contest trajectories. To avoid confusion, we have removed this contest from the supplementary material.

2. Line 27-28: The wording of this could be made clearer.

In order to comply with journal formatting requirements, we reorganised content across the abstract, the introduction, and the discussion – the same point is now made in lines 202-205, hopefully in a clearer way.

3. Line 67-68: You could discuss the idea of "punctuated equilibrium" from evolutionary biology (see [1]) which also has been used in technological innovation research from economics (see [2] and [3]).

We thank the reviewer for pointing this out, and now do discuss this in lines 208-210.

4. Line 69-70: "all entries bar the first contained at least one line of code from a previous submission", but in Figure 3 of the extended data there appears to be almost a third of the entries that have "zero new lines".

The entries that contain zero new lines in this context are the entries that have not introduced any new lines to the contest total, such that all the lines in a 'zero new lines' entry originated from previous submissions but have been combined or reordered in a novel way – by definition these entries must have more than one line of code from a previous submission.

5. Line 73-74: Before the number of novel lines were reported in terms of percentages, then it changes to number of lines of code. It would be helpful to also include the percentage.

Calculating the percentage of novel lines within an entry, and comparing this novelty across entries is relatively easy, but with respect to the code shared between entries, calculating and comparing percentages is not straightforward, because the percentage will depend on the number of lines each entry consists of. There is no common total number of lines since each entry has a different length. If two entries have, for example, 10 lines of code in common, these ten lines could represent 50% of one of the entries, and just 10% of the other one. This issue is amplified if we are interested in comparing across contests, each characterised by different problems and different solution sets. With this in mind, we think that the raw number of lines offers a clear means of comparison.

6. Line 83: Should it say "... after the current leader was no longer the leader"?

In an attempt to show that entries preferentially copy the current leader, we are comparing the similarity to the current leader with the similarity to entries that are not the current leader. Since the contests are long and the solutions entertained by the population change over time, it would not be useful to compare, for example, a current entry submitted in the middle of the contest with entries submitted at the beginning of the contest, which will most likely be very different. To constrain the comparison to recent entries, while still avoiding noise, we chose to look at the set of entries that had been submitted since the current leading entry but that did not achieve leader status. Therefore we are comparing the similarity to the current leader with the similarity to the set of recent entries that are not leaders, finding that the similarity to the current leader is higher.

7. Line 100 and 103: Both correlation values are negative, but the paper states that the "reverse relationship was observed" for the second value. Are the values correct?

Here, we calculated the score increment as the absolute difference between the score of the current leader and the score of each entry. For entries that manage to take the lead, a high difference corresponds to a high

improvement in the overall score, while for entries that do not take the lead a large difference corresponds to a particularly bad score, much worse than the current leading score. Therefore, a negative correlation between similarity and increment in leading entries means that entries that are more similar to the current leader introduce smaller improvements. A negative correlation in non-leading entries means that entries that are more similar to the current leader diverge less from the current best (i.e. have a lower score difference). Thus, although the direction of the correlation is the same, the interpretation is different – for leaders low similarity is beneficial, while for non-leaders high similarity is desirable. We added clarification in line 107.

8. Line 121-122: Are the increments in performance normalized across competitions?

They are, and we have updated the values in line 127 and added details to clarify this in lines 305-308.

9. Line 126-127: Another analysis of the change in solution diversity in the MATLAB competitions showed that the diversity in the population of solutions dropped after a new leading entry [4].

We are grateful to the reviewer for the suggestion. Unfortunately, the cited reference does not discuss how diversity was measured, so it is unclear how diversity here relates to our measure of diversity.

10. Line 138: It would be good to know how many unique contestants contributed a line of code to the leading entries. This would ensure that the collective intelligence was coming from more than one contestant.

Great suggestion, we calculated that this figure is on average 19 (+/-9 SD), which supports the collective intelligence view. We added this new analysis in lines 170-173.

11. Line 262: Should this line be "the total number of species..."

The reviewer is correct, changed line 262 (now line 275).

REVIEWERS' COMMENTS:

Reviewer #2 (Remarks to the Author):

The authors have addressed all the comments are requests for revisions I made satisfactorily. I have no further comments or concerns with the revised version of the manuscript.

Reviewer #3 (Remarks to the Author):

The authors have satisfactorily addressed my comments.